# Coping and Social Resilience during the COVID-19 Pandemic: A Qualitative Follow-Up Study among Healthcare Workers in Norwegian Public In-Home Services

**DOI:** 10.3390/healthcare10122518

**Published:** 2022-12-13

**Authors:** Aud Johannessen, Anne-Sofie Helvik, Kjerstin Elisabeth Tevik, Kirsten Thorsen

**Affiliations:** 1Norwegian National Centre for Ageing and Health, Vestfold Hospital Trust, 3103 Tønsberg, Norway; 2Department of Public Health and Nursing, Faculty of Medicine and Health Sciences, Norwegian University of Science and Technology (NTNU), 3103 Trondheim, Norway

**Keywords:** capabilities, crises, distancing, hygienic routines, mental health, properties, stress, resources, virus, work experiences

## Abstract

Background: Healthcare workers (HCWs) are central and serve in the frontlines when epidemics threaten public health. Thus, certain communities may be hardest hit by these challenges. Interventions supporting HCWs are important, and to develop these, understanding their experiences is essential. Aim: To explore how HCWs in Norwegian public in-home services experienced work during the COVID-19 pandemic over time. Method: A longitudinal qualitative study with two data collections approximately one year apart (2021 and 2022) was performed. Individual interviews were conducted with HCWs. Results: The analysis resulted in six main themes: Changing everything, Redefining ‘necessary tasks’, Distancing and loneliness, Cooperation and coordination, More infections and fewer worries and Lessons for the future. These indicate capabilities and processes, how they are evolving over time, and outcomes. The first two themes focus on the first period of the pandemic, the next two on the ongoing intermediate period, and the final two cover the last period. Conclusion: The HCWs’ narratives have demonstrated their collective coping based on adaptive and transformative capacities. Further, they have enlisted experienced social resilience in their strategies for coping with the COVID-19 challenges.

## 1. Background

At the time of this writing, the most-recent statistics report 589,680,368 confirmed cases of COVID-19, including 6,436,519 deaths, reported to the World Health Organization (WHO) and a total of 12,409,086,286 vaccine doses administered [1]. The COVID-19 pandemic has affected all members of societies worldwide, putting healthcare systems at all levels under pressure and testing their resilience. The WHO points out that healthcare workers (HCWs) represent one of the six building blocks of health systems [2]. Moreover, they are central to a health system’s ability to respond to highly significant public health challenges such as outbreaks of a pandemic. As personnel working in the frontlines are often hit hardest by these demands, interventions supporting HCWs in various ways are necessary to strengthen health systems’ resilience [3].

To develop effective interventions to support this group, several recent reviews have focussed on HCWs’ experiences during the COVID-19 pandemic and its impact on their well-being and mental health. Most of these reviews refer to psychological scales used to provide quantifiable information on HCWs’ well-being and mental health [4,5,6,7]. In addition, a scoping review including qualitative studies pointed out that this pandemic has presented HCWs with a number of challenges that have affected them and their experiences of needs for support [8].

In Norway, the signs of a global pandemic were perceived and rather clear by 30 January 2020, and by the end of February, the first patient to test positive for the coronavirus was hospitalized [9] (p. 88). On 12 March, the most radical peacetime steps were put into place [9] (p. 124), and Norway’s Health Directorate legally regulated the lockdown of the country’s workspaces and schools, requested that people remain at home, and advised them to quarantine there if they fell ill and to stay away from vulnerable persons and be tested for COVID-19 [10]. Several additional regulations were implemented after a steep increase in the number of severely ill patients in Norway. The testing, rules and regulations seemed to be somewhat efficient in reducing the spread of the virus among Norwegians [9] (p. 306), although wide variations in COVID-19 infection rates were identified between municipalities, with smaller municipalities faring better in the early phase of the pandemic.

Most studies on the effects of the pandemic on HCWs’ health have focussed on negative outcomes, especially for mental health, reporting increased levels of distress, depression, anxiety and sleep disturbances [11,12]. However, while two of every five HCWs have reported such problems, three of five have not reported elevated stress, even during extraordinarily challenging working conditions and societal transformations caused by the pandemic [11,12]. Overall, they have been found to be coping (managing stress and demonstrating resilience [13] (pp. 53–79). The common understanding of resilience is that it is a multifaceted concept with complex dimensions and relates to the ability to bring about positive outcomes when faced with external adversities [14]. Resilience has mainly been considered a psychological concept that has been studied as an individual capacity [15]. As pointed out by Fan and Lyu [16], in the last decades, resilience research has gradually become ‘a hot topic’ in many disciplines including developmental psychology, mental health, management and environmental research. The concept has been expanded into the level of ‘social units’ and communities, summarized by the concept of social resilience. The social unit demonstrating resilience can be studied at individual, household/family, organization, community and system levels, depending on the research focus [14]. Furthermore, social units may include multiple social sub-entities and/or interact with other social entities. The concept of social resilience encompasses a time perspective and includes both the abilities to cope with unexpected risk events in advance [16] (p. 2) and to manage their consequences during and after the event. Two specific properties for social resilience have been identified, risk sensitivity and regenerative properties, and communities and social institutions need both to prepare for emerging risks and to manage and overcome events [14]. Resilience can be understood as an attribute (like a capacity), a process and/or an outcome associated with successful adaptation to—and recovery from—adversity [17].

Keck and Sakdapolrak [18] (p. 5) have suggested that social resilience capacities have three dimensions: coping, adaptive and transformative (CAT) dimensions. Coping capacities entail the ability of social actors to cope with and overcome all kinds of adversities. Adaptive capacities involve one’s ability to learn from past experiences and adjust to future challenges in everyday life. Transformative capacities address the ability to craft sets of institutions that foster individual welfare and sustainable social robustness for future crises. Thus, social resilience capacities are seen as more than coping with, adapting to and recovering from hazardous events; they also involve the capacity for the transformation of social systems to another state. Social resilience is, therefore, conceptualized as a collection of proactive as well as reactive capacities for transformation [19].

The social resilience process includes the social mechanisms at the system level and may involve decision-making and the use of resources, features and community action [14]. The outcome related to social resilience may be the experiences of the COVID-19 pandemic actions in evaluations of how the immediate challenges were handled as well as the (greater) preparedness for future hazards.

Based on previous research related to hazards including COVID-19 [14,16], it is recommended that future research regarding social resilience include personal and organizational perspectives, as well as the cultural context, to better understand how social resilience is developed and preserved. Applied to the COVID-19 pandemic in Norway, this study of HCWs in public in-home services focusses on their work experiences over time including capabilities, processes and outcome aspects.

## 2. Aim

The aim of this study was to explore how HCWs in Norwegian public in-home services experienced work during the COVID-19 pandemic over time.

## 3. Method

### 3.1. Design

Qualitative individual interviews [20] (p. 110) were conducted with HCWs in Norwegian municipalities’ in-home services during the COVID-19 pandemic. A longitudinal qualitative study with two data collections approximately one year apart (2021 and 2022) was performed. Individual interviews were conducted to capture experiences, coping efforts and strategies over time. Both interviews covered current experiences as well as retrospective accounts of earlier reactions. The follow-up was chosen to obtain detailed and precise information about changes that took place throughout the interviewees’ pandemic work experiences.

#### 3.1.1. Participants

For heterogeneity, we contacted 10 managers of municipalities’ in-home services and two professors of old-age health courses who were teaching postgraduate students at two universities. They were asked to recruit HCWs in the municipalities’ in-home services, and all 12 recruited participants by phone or in face-to-face contacts. A total of 13 municipalities and universities represented Northern, Central and Southern Norway, and 18 participants were recruited, three of them men. The participants comprised 13 HCWs and five university students also working in in-home services; they ranged in age from 25 to 64 years old, and their experience ranged from 2 to 32 years working in-home services. In regard to occupations, 12 were nurses, two were occupational therapists, and four were nursing assistants. During the period in which the interviews were conducted, all 18 had experience with COVID-19 among patients, caregivers or in their own families. At the time of the one-year follow up, one participant had changed workplace, and two did not respond to a phone call to request a follow-up interview (one male nurse and one female nursing assistant).

#### 3.1.2. The Interviews

Individual qualitative interviews were conducted in two phases. The first took place at the inclusion of the participants in 2021 and the second at follow-up approximately a year later (2022). This relatively short interval between interviews was chosen because significant aspects of experiences of their work situations during the COVID-19 pandemic may have changed rapidly and may have varied in different municipalities. All interviews were conducted by the first author (AJ). Because of pandemic-related restrictions, interviews were conducted by telephone or FaceTime, based on participants’ resources and preferences, and at the most convenient time as chosen by each participant [20,21] (p. 514; p. 131). All interviews were audio recorded and transcribed verbatim within two weeks by a professional typist. The interviewer performed a quality control check by listening to the audio recording while reading each interview.

Both the initial and follow-up interviews were based on the same interview guide with five broad, open-ended thematic questions focussing on participants’ experiences of working in a municipality in-home service during the COVID-19 pandemic. The questions were developed for this study and are listed in Table 1. Depending on the interviewees’ replies, the aspects and ideas they raised led to the interviewer asking additional questions to clarify responses or to gain additional information. At the follow-up interview a year later, an additional question was included on the interview guide, with the aim of exploring in greater detail and more specifically what had happened since the first interview. The participants were asked the following summarizing open question: All in all, how have you experienced the COVID-19 pandemic at your work?

As shown in Table 1, the first and third questions were similarly formulated and raised the topic of the working situation and its description. The second question referring to collegial collaboration is more specific, and the last two explore the situation for the care recipients. Both the first and second interviews had a time perspective to cover the interviewees’ present and former situations related to the pandemic. The second interview also invited reflections about the future.

##### Phase 1

In the first phase, 12 of the 18 interviews were conducted by telephone and six via FaceTime. The interviews lasted for 17 to 36 min (for a total of 423 min) with a mean of 24 min.

##### Phase 2

The 16 interviews in this phase were conducted by telephone. They lasted for 14 to 26 min (for a total of 274 min) with a mean of 18 min.

### 3.2. Analysis

Data were analyzed using manifest thematic analysis as described by Braun and Clarke [22] (p. 83). This method has been used to identify, analyze and view both coincident and divergent patterns or codes in the total qualitative data material [22] (pp. 87–93). The process began by listening to the audio files several times to become familiar with the data material, which was then read, and initial thoughts about what was important regarding the capacities, processes and outcomes of COVID-19 took shape. Interesting aspects from each interview were identified and labelled with descriptive codes, and codes with similar meanings were organized into the same theme. Some of the themes were used in several different codes. A review of the transcriptions and codes was performed to determine whether each code reflected the meaning of the text in the transcriptions. The codes were then organized as overall themes, and topics not relevant to the aim were excluded from further analyses. Moreover, quotations that illustrated a theme in a nuanced way were selected and added to the results. Authors KT and AJ had principal responsibility for the analysis, but the process was discussed among all the authors. After several rounds, it resulted in the names that are presented in the results section.

We examined experiences, emotions and events that differed over time and noted the capacities, process and outcomes individually and collectively. Retrospective information was collected at both the first and second interviews, thereby being responsive to the situation and the phenomenon under study [21] (p. 464).

A tentative overriding theoretical concept was ‘coping’. This was originally launched as an individual psychological concept, denoting the abilities to overcome stress and strain and, thereby, demonstrate individual resilience. The analysis of our study convincingly demonstrated that the HCWs’ efforts to manage the impact of the pandemic were understood and described as collective. The concept ‘social resilience’ then came to the forefront as the most relevant theoretical notion. The process of defining concepts theoretically was discussed by all the authors throughout the analysis process.

## 4. Ethics

The study followed the ethical principles outlined in the Declaration of Helsinki [23] and was presented to the Data Protection Services, which determined that the Norwegian Medical and Health Research Act did not apply (number: 335429). Thus, the study did not require approval from the Regional Committees for Medical and Health Research Ethics. Informed consent was collected from the participants after they had received verbal and written information about the study and before the interviews took place.

## 5. Results

The analysis resulted in six main themes: Changing everything, Redefining ‘necessary tasks’, Distancing and loneliness, Cooperation and coordination, More infections and fewer worries and Lessons for the future. The themes indicate capabilities, processes that evolved over time and outcomes. The first two themes focus on the first period of the pandemic, the next two on the ongoing intermediate period, and the last two cover the final period (see Table 2). Variations were identified between the included municipalities in regard to the degree of contagion, the organization and the actions taken; however, the analysis did not reveal systematic differences in the experiences of the respondents from the included municipalities. No differences were found in trends by the collection of the data material using telephone and FaceTime. Quotes from the participants are labelled as the number of the participant and the phase. Thus, 12, 1 indicates a comment from participant 12 in phase 1 of the study, and 12, 2 indicates a comment made by participant 12 in phase two.

### 5.1. The First Period of the Pandemic

#### 5.1.1. Changing Everything

The municipalities were caught by surprise when the COVID-19 restrictions were first introduced in Norway on 12 March 2020. The participants used the following expressions, among others, to describe the sudden and abrupt changes in practice: ‘turned around’, ‘upside down’, ‘an abrupt change’, ‘a merry-go-round’, ‘totally new working conditions’, ‘it changed everything’ and it felt like ‘a strange world’. The participants experienced surprise, insecurity and stress; some mentioned fear, and one commented, ‘I have been scared’ (17, 1). Another said, ‘In a way, we became frightened and a little bit panicky’ (16, 1). This feeling was shared even by those who did not have a single care recipient in the municipality who was infected. It took some time to develop and implement new systems for care. A participant described this period as ‘a steep learning hill’ (10, 1). Another stated, ‘The leaders had a lot of meetings about how we should meet the challenges. Gradually, we implemented new systems, with infection protection and rules 11, 1′. The leaders’ ways of communicating were essential to reduce fear and insecurity. A participant stated, ‘We received a lot of good information emphasizing that they (the health administration) had control. Gradually, we felt that the situation was safe. All the time we got instructions in how to do it’ (16, 1). This participant mentioned the leading infectious-disease doctor in the municipality in particular, who used Facebook and the internal municipalities’ home pages to disseminate updated information. Several participants reported that they had confidence in the information they received: ‘We relied on and trusted the advice and rules. So, for me personally, I felt no fear’ (11, 1).

If participants described being scared, what did they fear? No one mentioned being afraid of being infected with the coronavirus themselves, but they were deeply afraid of being infected without having symptoms and unknowingly passing the virus to their care recipients. Some said that they were always worried: ‘Am I infected? My son? Am I infecting some care recipients?’ (12, 2).

In nearly all the municipalities, HCWs lacked equipment needed to protect themselves from infection. ‘We had no protective glasses, no protective coat, no gloves, and we had only one mask a day to use’, reported one participant (6, 1). She continued, ‘Lack of everything (…). Absolutely everything we needed has been difficult to get’ (6, 1). The need for a quick reorganization became a force for rapidly abandoning former rules and introducing new practices. One participant mentioned that the rule about written acceptance from care recipients (signatures for application of equipment) was replaced by verbal acceptance by phone (7, 1). The personnel introduced timesaving solutions—less time in face-to-face meetings, more electronic contact, interactions and applications. A participant noted, ‘New solutions were enforced (…). That is okay. It is the way it is. Things have been postponed’ (7, 1). The shared situation was summarized by another participant who described that they had all been assigned many more tasks but fewer personnel to handle them (9, 1). The expansion of work tasks is highlighted in this sentence: ‘The infection rate has increased gradually, and all the capacity has gone to that’ (9, 1). Furthermore, some municipalities prepared for ‘the worst-case scenario’. The new working situation was described as ‘hectic’. A participant working under the new schedule (7 days on and 7 days off) reported: ‘Being in a high gear, not able to relax, always worried’ (6, 1). However, she added, ‘For me, it has been positive because my knowledge and abilities have been needed’ (6, 1).

The intensive work and effort have escalated, and the extra burden is felt. ‘We see that most of the personnel have a short fuse. (…) I don’t see much joy these days!’ (6, 1). Too few staff, not enough personal protective equipment and very little praise and encouragement have taken their toll.

Feeling constantly worried, one respondent asked, ‘Is this good enough help from the municipality? Is what we do satisfactory?’ (11, 1). At the same time, HCWs seemed to accept the restrictions and the work tasks they were able to perform under the new circumstances, while relying on the advice and rules provided by the authorities. This was described by a participant who stated, ‘It is just the way it is. You just have to stay and manage the best you are able to’ (11, 1).

#### 5.1.2. Redefining ‘Necessary Tasks’

The participants described how all the basic functions of their work were retained but reduced to the ‘necessary tasks’. The basic principles of the services were redefined and *visits* were no longer ‘basic’, as described by this participant: ‘We have reduced many of the visits. Not all are really necessary’ (10, 1). Care recipients and personnel alike could, for varying reasons, wish to reduce the number of visits from the HCWs. For the personnel, this emerged from a necessity to reduce workloads, and for older care recipients and their families, it was to reduce their risks of becoming infected. Several care recipients lost their usual visits without any choice. A participant described this change, saying, ‘We omit visits to some who get assistance with very simple tasks’ (17, 1). Help putting on support stockings, showering and making sandwiches, which the care recipients managed themselves, were mentioned. However, not providing assistance with support stockings could have deleterious effects, as this participant explained, ‘It resulted in wounds and edema, and many older care recipients became more reduced (in daily functioning)’ (6, 1). Another added, ‘We had left just those older care recipients who were in need of basic (physical) health assistance (…). Unnecessary visits are those mostly for mental health and social needs’ (3, 2). At an unplanned visit to an older care recipient who usually received phone calls, a HCW discovered that the person was ‘in bad condition’ and needed hospital treatment (10, 1). As shared by the participants, downscaling visits could result in serious consequences for the care recipients.

### 5.2. The Ongoing Intermediate Period

#### 5.2.1. Distancing and Loneliness

Many older care recipients longed for contact, as illustrated by one participant: ‘When we say that we have to keep (our) distance, some reply, ‘It does not matter. Just come closer and touch me!’ (10, 1). The same participant mentioned a situation where an older care recipient threatened to call for an emergency ambulance: ‘When we arrived, her only need was for a glass of water’ (10, 1). However, this participant added that other older care recipients were deeply frightened, some ‘out of their wits’ and stressed by the necessity to maintain social distance.

For HCWs, keeping a distance between themselves and the care recipients made providing good healthcare assistance more difficult. One explained this further by saying, ‘Occasions for observations are reduced when you must stay at a distance. For example, when you can’t sense if the hand is dry, cold or clammy. Then, the whole evaluation is based on what the older care recipient says’ (10, 1). This participant continued, ‘We are much more concerned when it comes to hygiene (…). Some colleagues just neglect the psychic situation. It is not visible’.

Many participants mentioned loneliness among the older people. One stated, ‘There seems to be a hunger for contact and small talk’ (11, 1). Isolation was increasing over time and affecting the care recipients, as described by this participant: ‘When the day-centre was closed, voluntary visits stopped, and support contacts resigned, then what they had for social contact, which had also been a relief for their families, came to a stop. Life became empty and dreary (…). When you don’t see the smile, that is bad!’ (14, 1). A client had remarked, ‘We live like prisoners’. The older care recipients’ needs for conversation could become overwhelming, as described by this participant: ‘When you visit an 85-year-old woman and see that she has a great need for talking to someone, and then you really have no time! I have felt that very much the last year. The result is that you stay to talk and get delayed, and so it continues’ (6, 1).

The HCWs themselves could also long for small talk. One participant described a short phone call from a colleague who needed information that turned into an hourlong private chat (10, 1).

Another participant remarked, ‘Older people can’t stay a long time being depressed and lonely before their physical health is affected. You see that people just give up (…). The physiotherapist and occupational therapists have ended their visits in certain periods. These may be just as important as our visits’ (6, 1).

‘The older people now are more open about being lonely. It goes with the situation’ (6, 2). Loneliness is attributed to a shared situation and not as an individual failure or weakness.

#### 5.2.2. Cooperation and Coordination

As mentioned above, the reorganization of the home services to align with the restrictions imposed by the COVID-19 pandemic had to take place immediately, but its implementation was modified over time. The dominating situation was that additional tasks were added, usually with fewer personnel. As the pandemic gradually spread, more people became infected; a system for testing for the virus was established, and later vaccinations were introduced. One participant pointed out that, during this process, various professions seemed to lose their specific specialist perspectives, and the boundaries fences between them were lowered. The force of necessity promoted an orientation towards ‘finding new solutions’ together (7, 2). At the same time, personnel for the provision of in-home care services had to practice a process of continual evaluation. The same participant stated, ‘It has functioned surprisingly well. We do find solutions!’ Over time, ample equipment became available.

As more COVID-19 tests were introduced for additional age groups and, later, vaccinations, personnel resources were allocated to these new tasks. In some districts, staff were divided into two or more *cohorts* that did not share care for older recipients in order to reduce the risk of transmitting the virus. One participant commented, ‘The work became more laborious and time-consuming’ (8, 1).

It was apparent that, for the continuing process to work, an attentive and engaged leader needed to be present. One participant emphasized what the absence of a leader who was present meant: ‘We were in a special schedule from April to May. We missed getting support from a leader, telling us that we did a good job. We felt very much alone!’ (6, 1). As this remark underscores, to be seen, appreciated, confirmed and praised for their hard work was found to be a critical element for the HCWs’ job satisfaction and endurance.

### 5.3. The Last Period

#### 5.3.1. More Infections and Fewer Worries

A shared experience among the participants over time was that, although the number of infections increased, there were fewer worries (10, 2). One explained this by saying, ‘They find that they have got better routines and these are more established’ (8, 2); another said, ‘The shoulders are lowered’ (6, 1). The situation is seen in contrast to the first stage, when everything was new, unexpected and continually changing. Fear was reported to be decreasing among HCWs as well as the older care recipients. While more people may have been becoming infected with the virus, older care recipients seemed to have fewer questions and were less insecure and less afraid of contracting it. One participant stated, ‘They are more trusting towards the HCWs and the virus protection (measures)’ (7, 2). Another commented, ‘This reduced the workload’ (6, 2).

Nearly all the interviewees mentioned that, after the first two hectic phases of the pandemic, they were better able to cooperate with their colleagues, were more attentive and provided more assistance. One participant described this by saying, ‘We don’t stand alone. If we wonder, we have someone to rely on. I feel we are more helpful and experience more confidence. I feel no problems. We are focussed on finding solutions. The positive atmosphere is recognized by others. The vicars told us that it was nice to work with us’ (10, 1).

The schisms mentioned did not appear within the group but in other contexts. One participant mentioned conflicts with the hospital staff regarding the admittance and transfer of older care recipients back to the municipality. At the same time, the heavy workload over time and the increasing number of infections in the community resulted in even more sick leaves. A participant commented about this, ‘The burden is greater now, and the main problem is to find vicars. Sometimes seven or eight nurses are missing in a team. Then you feel the strain’ (10, 2).

#### 5.3.2. Lessons for the Future

Over time, with new organizational structures, new rules and new practices, the HCWs have figured out which factors seem to function better than the previous regulations and may be transferable to a normal situation. They have also demonstrated proactive capacities. The division of the staff into ‘cohorts’ is seen over time to have several advantages. According to one participant, ‘It became easier to cooperate and to share information. It has resulted in much better continuity in in-home nursing care (fewer persons go to the older care recipients)… A very positive experience and we really appreciate and want to preserve the continuity in home nursing’ (8, 2). The small, close-knit ‘cohort group’ is seen as more competent and resourceful. Summarizing, one participant expressed it this way, ‘We have been a very small and well-integrated group that has managed to meet the challenges. We see that we do a better job, and then the satisfaction increases. We do experience a strong cohesion in the working group’ (16, 2). Another participant remarked that the HCWs have discovered the importance of other professions and mentioned the housekeepers. The necessity of intelligent, attentive and present leadership was emphasized. Staff meetings via video were seen as a time-saver; one participant explained, ‘We have got a more flexible workday’ (11, 1).

One of the most decisive lessons they have learned relates to better hygiene. The participants received information from the media about how strict hygiene procedures result in fewer other infectious diseases such as pneumonia and influenza. They see the need for better hygiene in the in-home services. A participant commented, ‘… I think hand-shaking will be markedly reduced in the future, and we will be more alert’ (6, 1).

However, there is doubt whether some of the valued changes will be continued as the situation returns to normality. In particular, needs for meetings, small talk, and face-to-face contact with colleagues will probably result in more close interaction with more people. Some of the rules have already weakened or been terminated, reported a participant who further pointed out that there are nearly no COVID-19 rules left. Some previous rules may be less strict; others are vacillating and thus unclear. Some HCWs have stopped wearing masks in certain situations, while others are reminding people that masks should remain in use for the time being.

## 6. Discussion

The participants’ experiences are narrated in stories centred around particular trials involving intensive, stressful and challenging work, as well as some fear, in cooperation and helpful interaction with colleagues. Narratives about persistent internal conflicts, aggression and bitterness among HCWs are mainly lacking. Also, mostly absent are stories relating criticism of politicians, the government, and most national health administrators and local leaders. A repeated narrative involves the early lack of equipment, plans, and national and local preparedness before ‘everything changed’ abruptly. The high level of stress and heavy workload could sometimes lead to situations where people were reacting irritably and taking sick leave. However, the overriding story of their work situation is about how everyone made extra efforts and found new solutions together with more cooperation and fewer boundaries and barriers between the professions than earlier. The sentence ‘We don’t stand alone’ summarizes their shared working together in a productive, constructive and collective atmosphere.

Thus, the narratives emphasize the collective efforts put forth throughout all the stages of the COVID-19 pandemic based on communication, cooperation and coordination. As a result, the participants described having experienced better cohesion in work groups, and the allocation of HCWs into ‘cohorts’ has strengthened feelings of ‘togetherness’, of sharing the challenges and tasks. The participants’ stories show that the HCWs have demonstrated collective coping based on adaptive and transformative capacities. They have exerted social resilience in their strategies for handling the many challenges presented to them by the COVID-19 pandemic.

The new organization has resulted in far better continuity in the contacts with the older care recipients. Discontinuity and rotation of staff have been recognized as a problem of the organizational structure of Norwegian in-home services for older people for many years [24]. One HCW condensed the participants’ opinions of the whole process by saying, ‘It has functioned surprisingly well.’ Many participants related that the HCWs have functioned together even better than they did before the pandemic because of the force of necessity and the need for immediate action. We found that, even in the later stages of the pandemic after a long, stressful period with strict regulations and isolation rules, there were no signs of accumulated criticism. The participants expressed confidence in and reliance on their leaders’ rules and advice. They have trusted that the recommendations and strategies they were given were the best available at each particular stage of the pandemic. Another study also pointed out that personal and organizational resilience during the COVID-19 pandemic might have influenced HCWs’ experiences and helped them develop alternative ways of working and collaborating [25].

Are there factors in Norwegian society that could support the study’s results of an approach of confidence and cooperation among HCWs? When it comes to the government’s efficiency in handling the COVID-19 pandemic, Bloomberg’s COVID Resilience Ranking [26] placed Norway at the top of the list, with New Zealand and Switzerland in second and third place respectively. It is reported that Norway is ‘one of the only places’ among the 53 included countries to have combined success in the initial stage of the pandemic with the recovery stage. The participants evaluated the advice and rules they received not as ‘the absolute best’ but as the ‘best’ considering the knowledge, experiences, equipment and legal and societal conditions present at that time. They acted accordingly: ‘It is just the way it is. You just must stay and manage the best you are able to!’

A unique Norwegian expression—dugnad—was used repeatedly by the authorities and government in media and information bulletins to gain acceptance and support from the population during the pandemic. It means doing something collectively out of a feeling of wanting to contribute to the public good—to the collective—in the best possible way. An indicator of this ‘dugnad approach’ is that Norway has a large and vital voluntary sector that is also stimulated and supported by the government [26,27]. A conclusion is that the experiences of the HCWs in the in-home services—their coping strategies, confidence and cooperation—are deeply integrated in other central values of Norwegian society and have been supported and sustained by these during the pandemic.

In the initial phase, the HCWs stated that the organizations and they themselves were totally unprepared for the effects that the COVID-19 pandemic would have on their work of providing in-home service to older vulnerable adults. They were concerned about the situation for those they were caring for. The HCWs described that they experienced stress and fear, but predominantly not for their own physical and mental health or well-being. Later on, in the process of coping with the pandemic and its many effects, they were either not mentioning or were downplaying stressful concerns or consequences regarding their own mental health. Thus, our results are not quite compatible with the scoping review’s report of COVID-19 affecting HCWs’ mental health negatively [8]. We may suggest that the HCWs’ ways of handling and managing stress individually and collectively in cooperation with their organization, the municipality and the Norwegian authorities and society, as well as the prosocial cultural ‘dugnad’ approach, have limited most negative effects on their health and well-being.

### Strengths and Limitations

With the significantly heterogeneous sample of HCWs providing in-home service to older adults during the COVID-19 pandemic, we were able to capture their varied experiences and situations and the complexities related to work and life [28] (p. 305). It is hoped that the participants will find a ‘fit’ between their experiences and the research accounts—what Charmaz [29] (pp. 182–185) called resonance.

To establish trustworthiness, during every stage of the study, we focussed on securing credibility and trustworthiness of the results [21] (pp. 557–559). The interviewer posed new questions during the interview to obtain more information [30] (p. 151). The transcripts were controlled by the interviewer, who listened to the tapes repeatedly to ensure that they were correctly understood and interpreted. Significant excerpts from the interviews were compared and analyzed, and the most representative excerpts were eventually included in the text under the themes. We adhered to the criterion described by Glaser and Strauss [31] (p. 289) that sufficient detail and description should be presented as empirical evidence for the results. We also focussed on the criterion for evaluating credibility: establishing logical links between the gathered data and the argument and analysis [21] (pp. 557–559).

Another strength of the study is that both interviews covered experiences today as well as retrospective accounts of earlier reactions. Two interviews were chosen in order to gain detailed and more-precise information about changes taking place throughout the COVID-19 pandemic work situation. The interviews also provided understanding about HCWs’ experiences and reflections during these changes throughout the pandemic and their impacts on their working situation.

However, to examine and compare the strategies and resilience expressed by the Norwegian participants, other studies in other contexts are necessary [21] (p. 164).

Limitations of the present study are its small sample size and the time period before the first interview, which was conducted in 2021. Another limitation could be that the leaders in the municipalities in-home services recruited the participants; this may have led to the selection and inclusion of participants who coped relatively well with the COVID-19 pandemic and its impacts on HCWs and their work situation and, thereby, to relatively more-positive responses. On the other hand, there were two teachers at two universities who did not know the participants they recruited. What lessons were learned and persisted after the pandemic restrictions were lifted would be interesting questions for further research.

The COVID-19 pandemic precluded face-to-face interviews. All interviews were, therefore, conducted by telephone or FaceTime as methodological approaches required by necessity. We still consider the information from all interviews, lasting a total of 697 min, to be valuable, informative and dependable.

The study included open questions concerning reactions to the COVID-19 pandemic, which is seen as a strength. However, this approach may have been instrumental in the release of fewer negative emotional reactions related to in-home service to older adults during the pandemic. Thus, further research might compare these findings to those of quantitative research using inventories designed to measure symptoms of anxiety or emotional stress due to in-home service during the COVID-19 pandemic.

## 7. Conclusions

The overriding narrative in the results of the HCWs’ work situation in Norwegian public in-home services was about how everyone made an extra effort and sought to find solutions together, with more cooperation and fewer boundaries and barriers between the professions than before the pandemic. The results running throughout the participants’ interviews show that the HCWs have demonstrated collective coping and use of adaptive and transformative capacities. Further, they have exerted social resilience in their strategies for handling the many challenges that came with the COVID-19 pandemic. They also stated that the reorganization has resulted in far better continuity in contacts with the older care recipients.

## Figures and Tables

**Table 1 healthcare-10-02518-t001:** The interview guide and its five main questions.

How are you experiencing your work situation during the COVID-19 pandemic?
2. How are you experiencing collegial collaboration at work under the COVID-19 pandemic?
3. Is the COVID-19 pandemic influencing your work situation?
4. What kinds of experiences have you had in the course of providing services to older recipients in the municipality in-home service during the COVID-19 pandemic?
5. How do you think older adults are experiencing the situation during the COVID-19 pan demic?

**Table 2 healthcare-10-02518-t002:** Overview of the healthcare workers experiences of public in-home services throughout the pandemic.

The First Period of the Pandemic	Changing Everything
	Redefining ‘necessary tasks’
The ongoing intermediate period	Distancing and loneliness
	Cooperation and coordination
The last period	More infections and fewer worries
	Lessons for the future

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
