# Peer review of "Coping and Social Resilience during the COVID-19 Pandemic: A Qualitative Follow-Up Study among Healthcare Workers in Norwegian Public In-Home Services"

_healthcare, 2022, doi:10.3390/healthcare10122518_

Round 1
Reviewer 1 Report
Please check the attached PDF file.

Author Response
Reviewer 1
Thank you for all your valuable comments and suggestions.
Thank you for this comment. We have made changes. Hopefully this is in line with the suggestion.
Hopefully our changes are in line with this comment now
Perhaps, the text was unclear, but the follow up-questions are described in the text. We can make further changes if necessary.
We have made changes in this section.
Thank you for this comment. We have mowed the sentence to the result section.
We can easily see that and have made changes.
Reviewer 2 Report
Overall comments for authors: The study is well thought through. However, there few steps that can improve the article.
There is a need for a thorough proofreading and some sentences should be well structured. Attention should also be paid to the spelling of words. e.g. focuses and not focusses
Background: the sentence in lines 108 to 110 is not clear. “Applied to the COVID-19 pandemic in public the in-home services focusses on their work experiences over…” It will read better without the article “the”.
Aim: the aim of the study should be refined. It could read as: “To explore how HCWs in Norwegian public in-home services experienced work during the COVID-19 pandemic”.
Participants; Line 132 -134 is clumsy. This should be rewritten. It will read better if the authors first state the total number of participants, and then break them down into the various groupings
Discussion: Line 441-442 : In this sentence “The high level of stress and heavy workload could sometimes lead to situations where people were reacting irritability and taking sick leave” is not clear.
The word “irritability” is not clear and does not fit the sentence structure.
Author Response
Dear reviewer 2,
Thank you for your comments.
- Language is not easy. So we have made changes in the whole article. It has been proof reading once more. Hopefully the resubmitted article is better.
- We have made the suggested change.
- Also, the aim.
- Thank you we have written it now in a way that was suggested.
- We can see that, and we have made changes.
Reviewer 3 Report
Thank you for the opportunity to review this important work on coping and social resilience among healthcare workers in public in-home services during the COVID-19 pandemic. This article shares insightful qualitative findings on the experiences of healthcare workers during various stages of the pandemic. The below comments are offered to support and strengthen this important work.
Abstract:
-The authors share that healthcare workers are hit hardest by the challenges of public health threats. I wonder if there is a better way to phrase this as some may say certain communities may be hardest hit (though many of these communities impacted by social determinants of health and structural factors are also healthcare workers). Perhaps changing the language to more precisely explain what the authors mean here.
Background:
-Great summary of the current statistics around the impacts of COVID-19 and vaccine progress. You could consider updating these to reflect the latest statistics.
-Excellent and succinct explanation of the resiliency capacities and dimensions. This may be very helpful for the readers.
Methods:
-It appears some results may be listed. Under participants, the description of the cohort may be better placed under the results section.
- It would he helpful to have a brief description of how healthcare workers were recruited by the 10 managers. For others who may be interested in doing similar studies, understanding if a convenience sample was used, what sign up processes occurred, and if participants were paid or given gift cards for their time would be helpful.
-I appreciate the thoughtful interview guide included. Were there any particular frameworks used or prior existing guides used to inform this guide? It would be helpful to list these if applicable. Was there any pilot testing done of the guide prior to use?
Analysis:
-How was consensus or agreement reached when deciding on final codes and themes? The authors shared that the process was discussed among all the authors and further detail would be helpful here.
Results:
-The results section is fascinating and the details shed light on many important aspects of coping and resilience during public health responses and recovery. Perhaps creating a table with quotes and the key words shared would be helpful for the readers to glean the most salient points.
-It may be helpful to have a visual that demonstrated your themes over the time periods described.
-Under lessons for the future, it appears this sentence may be missing a portion or perhaps it’s meant to be in quotes (“A very positive experience.”).
Discussion:
-This first sentence was a bit hard to follow. Are the authors meaning to say that the participants’ stories were centered around these particular traits?
-It was excellent to see the mention of making changes to work situations and organizational structure as this highlights the importance of organizational level change and its ability to impact overall wellbeing as well, in addition to personal resilience.
-In regard to the limitations, could social desirability bias be at play with these types of qualitative interviews? Additionally, are there any participants that may not have access to the resources necessary to participate (and therefore unable to participate) in the study (resources such as access to FaceTime or internet access)?
Conclusion:
-The final sentence was difficult to understand. If the authors could further clarify this sentence that would be helpful. Perhaps the word choice of “another result” may need to be changed as it may be confusing to read within the conclusion section.
Author Response
Dear reviewer 3,
First of all, thank you for your comments and suggestions.
- Thank you for this comment. We have made changes. Hopefully the text is better now.
- We have made changes.
- The suggestion about method I believe that it should be placed as it is in a qualitative research article, but of course changes can be made.
- We did not do a pilot testing of the interview guide, because we started with open ended questions, and then we followed up with further questions depending on the information we got from the participants.
- Hopefully the text is clearer no, but the process were discussed throughout the reading and writing process.
- In the presenting of the results, we can see that a table could have been made in the process, but we did not do that. To try to make an overview we have now made a table of the themes.
- In the discussion we hopefully have manage to write it better.
- Hopefully, we have manage to rewrite it better now in the limitation section.
- In the conclusion section we have managed to write it better now.